# Does improved interpreter uptake reduce self-discharge rates in hospitalised patients? A successful hospital intervention explained

Elise O'Connor[1], Vicki Kerrigan[2], Robyn Aitken[2,3], Craig Castillon[1], Vincent Mithen[2], Gail Madrill[1,4], Curtis Roman[5], Anna P. Ralph[1,2]*

1 Department of Health, Royal Darwin Hospital, Top End Health Services, Darwin, Northern Territory, Australia, 2 Menzies School of Health Research, Charles Darwin University, Darwin, Northern Territory, Australia, 3 College of Medicine and Public Health, Flinders University, Adelaide, South Australia, Australia, 4 Department of Health, Top End Health Services, Darwin, Northern Territory, Australia, 5 Aboriginal Interpreter Service, Darwin, Northern Territory, Australia

* anna.ralph@menzies.edu.au

## Abstract

### Background

Aboriginal language interpreters are under-utilised in healthcare in northern Australia. Self-discharge from hospital is an adverse outcome occurring at high rates among Aboriginal people, with poor communication thought to be a contributor. We previously reported increased Aboriginal interpreter uptake and decreased rates of self-discharge during implementation of a 12-month hospital-based intervention. Interrupted time-series analysis showed sudden increase and up-trending improvement in interpreter use, and a corresponding decrease in self-discharge rates, during a 12-month intervention period (April 2018—March 2019) compared with a 24-month baseline period (April 2016 –March 2018). This paper aims to investigate reasons for these outcomes and explore a potential causal association between study activities and outcomes.

### Methods

The study was implemented at the tertiary referral hospital in northern Australia. We used the Template for Intervention Description and Replication (TIDieR) as a framework to describe intervention components according to what, how, where, when, how much, tailoring, modifications and reach. Components of the study intervention were: employment of an Aboriginal Interpreter Coordinator, 'Working with Interpreters' training for healthcare providers, and championing of interpreter use by doctors. We evaluated the relative importance of intervention components according to TIDieR descriptors in relation to outcomes. Activities independent of the study that may have affected study findings were reviewed. The relationship between proportion of hospital separations among Aboriginal people ending in self-discharge and numbers of Aboriginal interpreter bookings made during April 2016-March 2019 was tested using linear regression. 'Working with Interpreters' training sessions were undertaken at a regional hospital as well as the tertiary hospital. Training evaluation comprised an

**Data Availability Statement:** This paper reports on data relating to Australian Aboriginal peoples. Open data is a cause of tension for Indigenous peoples

(see https://www.stateofopendata.od4d.net/chapters/issues/indigenous-data.html). Indigenous Data Sovereignty – that is, the right of Indigenous peoples to control data from and about their communities - has emerged as an important topic over recent years, raising "fundamental questions about assumptions of ownership, representation, and control in open data communities" (Indigenous data sovereignty: toward an agenda. editors: Tahu Kukutai, John Taylor. 20919. ANU Press). The data analysed in this paper were 1) Aboriginal language interpreter bookings data obtained from the Aboriginal Interpreter Service in Darwin, an Aboriginal-run, government-funded organisation, and 2) Aboriginal admissions data from the Top End Health Service. Our commitment to the data custodians of these organisations is that data will only be presented in collated form and only named study investigators will be permitted to access raw data. We committed to this to protect Indigenous data which might otherwise be used for data piracy. The approved ethics application states: Requirement: Detail any agreements with third parties to be given access to the data Response: Should this situation arise, third parties will not be given access to the data without written consent of the Menzies Project Leader Data access could be sought by contacting Professor Anna Ralph, Division leader, Global and Tropical Health, Menzies School of Health Research, Darwin, Northern Territory, Australia. t +61 8 89468647; anna.ralph@menzies.edu.au Dr Curtis Roman, Senior Director, Aboriginal Interpreter Service, Northern Territory, Australia. t +61 8 8999 8371; alternative +61 439175582; Curtis.Roman@nt.gov.au Michelle Matts, Manager, Ethics Administration, Menzies School of Health Research. t +61 8 8946 6687; michelle.matts@menzies.edu.au.

**Funding:** This study was supported by the Menzies School of Health Research Grants Scheme, and Improving Health Outcomes in the Tropical North: A Multidisciplinary Collaboration (HOT NORTH), National Health and Medical Research Council GNT1131932. APR is supported by a National Health and Medical Research Council fellowship (1142011). The funders had no role in study design, data collection and analysis, decision to publish, or preparation of the manuscript.

**Competing interests:** The authors have declared that no competing interests exist.

anonymous online survey before the training, immediately after and then at six to eight months. Survey data from the sites were pooled for analysis.

## Results

Employment of the Aboriginal Interpreter Coordinator was deemed the most important component of the intervention, based on reach compared to the other components, and timing of the changes in outcomes in relation to the employment period of the coordinator. There was an inverse association between interpreter bookings and self-discharge rate among Aboriginal inpatients throughout the baseline and intervention period (p = 0.02). This association, the timing of changes and assessment of intercurrent activities at the hospital indicated that the study intervention was likely to be casually related to the measured outcomes.

## Conclusions

Communication in healthcare can be improved through targeted strategies, with associated improvements in patient outcomes. Health services with high interpreter needs would benefit from employing an interpreter coordinator.

## Introduction

Effective communication between Aboriginal language-speaking patients and healthcare providers requires cultural respect and appropriate interpreter use. In many settings, use of professional interpreters has been shown to improve patient outcomes [1, 2]. However in Australia's Northern Territory (NT), where dozens of languages are commonly spoken [3], uptake of interpreters is low for people whose primary language is an Aboriginal language [4]. Poor communication compounds existing health disparities. Serious adverse outcomes for Aboriginal patients including death have been attributed to communication failures [5–7]. An adverse outcome considered a consequence of impaired communication is the high rate of self-discharge from NT hospitals, with resulting health costs to individuals, negative impacts on staff morale, and high health system costs due to associated unplanned re-admissions [8–10].

Previously we found that healthcare providers at Royal Darwin Hospital—the largest tertiary NT hospital—are often unaware of the need to use Aboriginal interpreters; that when need is identified, they face a convoluted bookings process and lack skills in working with interpreters; and that when an interpreter is booked, none may be available [4].

In response, Top End Health Services supported the employment of an Aboriginal Interpreter Coordinator at Royal Darwin Hospital for a 12-month pilot period. The study team supplemented this with additional activities including training sessions for healthcare providers about the NT Aboriginal Interpreter Service and how to work effectively with Aboriginal interpreters, and clinical championing of interpreter use, to produce a 'bundle' of interventions. The concept of 'care bundles' is common in hospital practice to improve the quality of care [11]. We conducted a quasi-experimental pilot study using interrupted time series analysis to determine effects of the intervention on interpreter bookings made (primary outcome measure), bookings completed and self-discharge rates by Aboriginal people (secondary outcomes), during a 24-month baseline period (April 2016 –March 2018) and a 12-month intervention period (April 2018—March 2019), and found that study activities were associated

with immediate and up-trending increases in Aboriginal interpreter bookings, and a down-trend in self-discharges [12].

The aims of this paper are firstly to explore the likely reasons for the improved interpreter uptake identified during the study period; specifically to determine which components of the intervention should best be invested in into the future, to sustain change. Secondly, we wished to further explore the likelihood of a causal association between study activities and the decrease in self-discharge rates which occurred during the study intervention period.

## Methods

### Design

This is an evaluation of a complex intervention [13] using the Template for Intervention Description and Replication (TIDieR) as a framework [14]. Each intervention component is described in the template according to what it comprised, who implemented it, how it was done, where, when and how much, whether modifications were made during the course of the study, and reach of each component (how well it was implemented). To assess 'Working with Interpreters' training, surveys of clinicians who participated were undertaken before, immediately after and 6–8 months after training.

The intervention, with main findings published elsewhere [12], was delivered during 2018–2019. In brief, the primary outcome, which measured healthcare provider behaviour, was the proportion of Aboriginal patients needing an interpreter for whom an *interpreter booking was made* in the intervention period (1 April 2018–31 March 2019) compared with the baseline period (1 April 2016–31 March 2018). Secondary outcomes were: proportion of Aboriginal interpreter bookings *completed* in the intervention compared with baseline periods, and proportion of Aboriginal admissions ending in self-discharge during the two periods. The study was conducted as part of a collaborative initiative called the 'Communicate Study' between Menzies School of Health Research, Royal Darwin Hospital and the NT Aboriginal Interpreter Service. The study complied with STROBE (Strengthening the Reporting of Observational Studies in Epidemiology) guidelines [15].

### Setting

The intervention package to improve Aboriginal interpreter uptake was evaluated at Royal Darwin Hospital (RDH), a 360-bed tertiary referral centre in the NT. One component of the intervention–'Working with Interpreters' training—was also conducted at Gove District Hospital, a smaller, regional facility located 1000km from RDH. However, evaluation of impacts on patients were confined in this paper to RDH. Around 100 Aboriginal languages and dialects are spoken in the NT [3]. Prior estimates indicate approximately 60% of Aboriginal people at RDH [4, 16] and in the NT [17] speak an Aboriginal language as their first language. Community consultation indicates that the majority would benefit from an interpreter in healthcare interactions [18]. We conservatively estimated that 50% of Aboriginal patient separations would benefit from an interpreter. RDH uses the offsite Aboriginal Interpreter Service which services a number of government agencies. Interpreters are available for face-to-face, telephone or audio-visual interpreting. The Interpreter Service also provides one 'rostered interpreter' to RDH on weekdays for four hours.

### Data and definitions

Top End Health Services interpreter bookings data for 1 April 2016–31 March 2019 were provided by the NT Aboriginal Interpreter Service. This database includes all requests made to the

service by RDH including ward, language and whether completed or cancelled (if so, cancellation reason). RDH separations data, used as a measure of inpatient healthcare utilisation, were obtained for all Aboriginal and Torres Strait Islander peoples for the same timeframe. Separations were classified as discharged or transferred, left against medical advice/discharge at own risk, died, unknown, other or change of care type. In this paper we use the term 'self-discharge' synonymously with 'left against medical advice/discharge at own risk'. Implications in the terminology that patients wilfully leave against advice convey incorrect assumptions (many patients misunderstand the need to stay). Torres Strait Islanders, admissions for dialysis, same-day procedures, private hospital, outpatient cardiology, borders and care provided in psychiatry units (where interpreter use is already high) [4] were excluded. After applying these exclusions, there were 21,633 separations among Aboriginal people in the two-year baseline and 10,919 in the 1-year intervention period [12].

To evaluate 'Working with Interpreters' training, participating clinicians at the two hospitals were asked to complete an anonymous, online survey before the training, immediately after and then at six to eight months to determine whether it was beneficial in improving their use of interpreters. To improve anonymity, survey data from the two hospitals were pooled for these analyses. The survey asked whether clinicians perceived improvements in their ability to determine the need for an interpreter, confidence working with an interpreter, clarity on interpreter bookings processes and estimated (self-reported) frequency of Aboriginal interpreter utilisation.

## Evaluation and analyses

The TIDieR framework was populated for each study intervention component: employment of an Aboriginal Interpreter Coordinator, 'Working with Interpreters' training, and championing of interpreter use. Timing of activities was compared to the interrupted time series analysis plots of interpreter bookings and self-discharge rates [12]. Other activities underway at RDH independent of this study that may have impacted study outcomes were ascertained and documented through discussions with other clinicians and researchers and members of the health services executive. Health service data custodians and analysts were consulted to determine whether capture of hospital separations data had changed during the period 2016–2019. Quantitative analysis was undertaken using Stata version 15.1 [19]. Statistical significance was defined as a p value of less than 0.05. Linear regression was used to examine the relationship between numbers of interpreter bookings made per month and self-discharge rates. Pre- and post- 'Working with Interpreters' training survey data were assessed using Chi-squared tests for trend.

## Cultural safety

A cultural safety lens underpinned delivery and evaluation. Cultural safety is a process empowering cultural identity and wellbeing, incorporating 'a systemic outcome that requires organizations to review and reflect on their own policies, procedures, and practices to remove barriers to appropriate care' [20]. A monitoring framework has been proposed by the Australian Institute of Health and Welfare to guide assessments of 'Cultural safety in healthcare for Indigenous Australians' [21], addressing how health care services are provided, how Aboriginal and Torres Strait Islander patients experience health care, and how culturally accessible health care is. The study responded to this framework by addressing *how health care services are provided* (whether Aboriginal interpreters are used) and *Indigenous patients' experience of health care* (reflected in self-discharge rate). Additionally, implementation and delivery included Aboriginal leadership, provided through the Aboriginal Interpreter Coordinator, the Royal Darwin Hospital Aboriginal Support Unit and the Aboriginal Interpreter Service.

**Ethics.** Approval was provided by the Ethics Committee of the Northern Territory Department of Health and Menzies School of Health Research (HREC-2017-3007 and HREC-2018-3245). Data used in this study comprised routinely-collected health service and interpreter service data, and anonymous surveys. Individual patients were not recruited into this study and therefore informed consent was not required.

## Results

### Interpreter bookings

Interpreter booking requests and hospital separations data among Aboriginal people admitted to RDH during the baseline and intervention periods are shown in Table 1. Bookings made for Aboriginal interpreters per quarter increased during the intervention period compared with the baseline period (Fig 1 and [12]).

**Table 1. Royal Darwin hospital separations (Aboriginal people) and interpreter booking requests during the baseline and intervention periods.**

|  | 2-year baseline period (N = 1333) | 1-year intervention period (N = 958) |
|---|---|---|
| **All admissions** | 21163 | 10919 |
| **Aboriginal interpreter bookings made** | 1333 | 958 |
| **Aboriginal interpreter bookings completed** | 761 | 607 |
| **Number of people estimated to need an Aboriginal interpreter** | 10582 | 5460 |
| **Proportion in need for whom interpreter booking made**[*] | 12.6% | 17.5% |
| **Proportion in need for whom interpreter booking completed**[*] | 7.2% | 11.1% |
| **Languages requested** |  |  |
| Yolŋu Matha | 478 (35.9%) | 315 (32.9%) |
| Murrinh-Patha | 244 (18.3%) | 158 (16.5%) |
| Kunwinjku | 126 (9.5%) | 116 (12.1%) |
| Kriol | 105 (7.9%) | 79 (8.2%) |
| Tiwi | 100 (7.5%) | 76 (7.9%) |
| Anindilyakwa | 65 (4.9%) | 67 (7.0%) |
| Burarra | 69 (5.2%) | 43 (4.5%) |
| Warlpiri | 50 (3.8%) | 35 (3.7%) |
| Other languages | 96 (7.2%) | 69 (7.2%) |
| **Location** |  |  |
| Inpatient ward | 870 (65.3%) | 467 (48.7%) |
| Telephone interpreter or audio-visual link[**] | 370 (27.8%) | 405 (42.3%) |
| Emergency Department | 26 (2.0%) | 29 (3.0%) |
| Intensive Care, High Dependency, Neonatal Intensive Care | 25 (1.9%) | 26 (2.7%) |
| Coronary Care Unit | 30 (2.3%) | 22 (2.3%) |
| Other locations | 12 (0.9%) | 9 (0.9%) |

[*]The proportion of Aboriginal patients estimated to benefit from an interpreter was set at 50% of hospital separations for Aboriginal people, where Aboriginal people were those coded as 'Aboriginal' or 'Aboriginal and Torres Strait Islander'. 'Torres Strait Islander and not Aboriginal' were excluded.

[**]hospital location for telephone and audio-visual interpreting not provided.

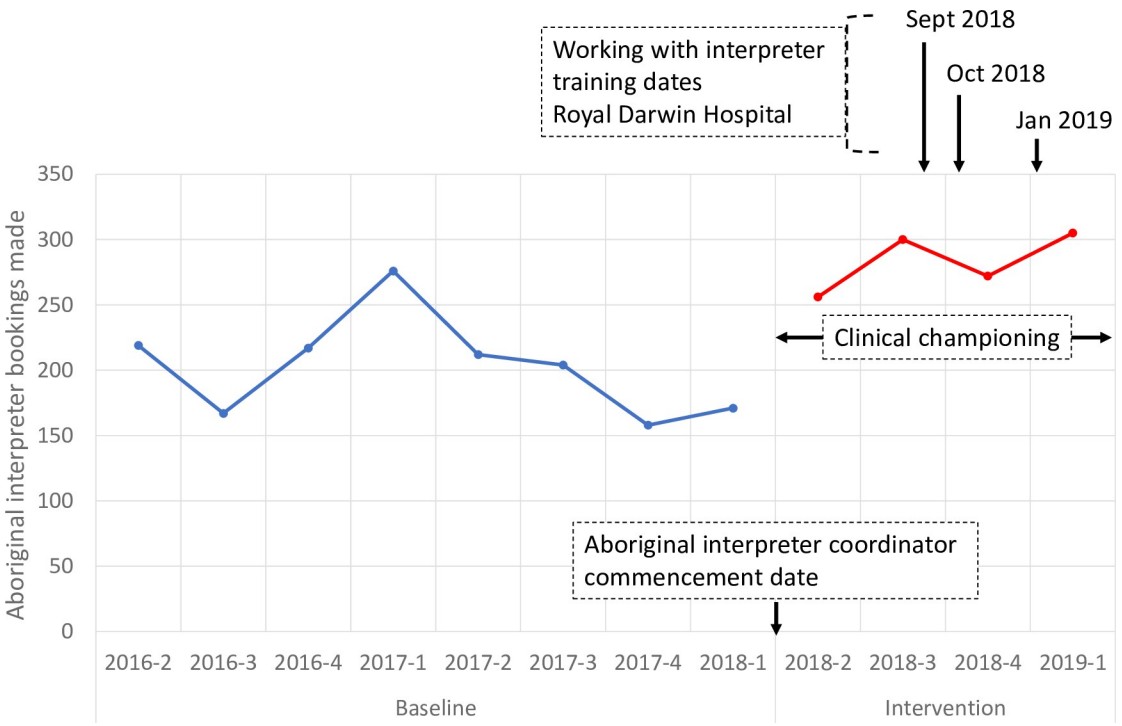

**Fig 1. Plot of raw data showing Aboriginal interpreter booking numbers made at royal Darwin hospital per quarter during the study baseline and intervention periods, annotated to show timing of intervention components.** Blue line: baseline period; red line: intervention period.

### Intervention component 1: Aboriginal interpreter coordinator

The study intervention period commencement date was the date of commencement of activities of the newly employed Aboriginal Interpreter Coordinator. This time point coincided with an increase in interpreter bookings, maintained throughout the intervention period (Fig 1). The Coordinator had training and experience as an Aboriginal Health Practitioner and Aboriginal Liaison Officer and was familiar with the health service. Tasks undertaken by the Coordinator are shown in Table 2. While the role was conceived as being a coordinator role to improve efficiency and ease of bookings, it was in fact realised somewhat differently, focussing more on Aboriginal staff support particularly, provision of mentoring for interpreters on assignment to the hospital, including helping them to navigate the hospital environment.

### Intervention component 2: 'Working with interpreters' training

'Working with Interpreters' training sessions were provided by the Aboriginal Interpreter Service in TEHS hospitals to 127 participants. At the tertiary hospital, sessions were conducted for the Emergency department and Surgical division doctors in September 2018 and October 2018, and for all new interns during orientation in January 2019 shortly before the intervention phase of the study ended (annotated in Fig 1). Surgery and Emergency are high-priority areas for interpreter use. Decisions about where to target training were also based on pragmatic factors, such as which teams had existing teaching rosters that could be utilised. Additionally, three training sessions were conducted for regional hospital staff in June 2018 (126 doctors and 1 nurse; Table 2); survey data from these sessions were pooled for the purpose of assessing the value of the training. 88/127 (69%) participating clinicians did the pre-training

**Table 2. Description of study intervention activities according to the template for intervention description and replication (TIDieR) checklist.**

| Item | Description | | |
|---|---|---|---|
| Name | The Communicate Study: A health systems intervention to improve uptake of Aboriginal interpreters at a tertiary referral hospital. | | |
| Why | An estimated 50% of hospital separations of Aboriginal people at Royal Darwin Hospital are for Aboriginal language speakers who would benefit from the use of an interpreter, but few get access. Ineffective communication about health matters including diagnosis, treatment and prognosis is associated with poor health outcomes, while interpreter use can improve outcomes. Systems changes are needed to support greater uptake of interpreters. | | |
| When | April 2018-March 2019 | | |
| Intervention components | **Employment of a hospital-based Aboriginal Interpreter Coordinator** | **'Working with Interpreters' training sessions** | **Clinical championing of interpreter use** |
| What | to address barriers to interpreter use. The Aboriginal Interpreter Coordinator was introduced at a Hospital Grand Rounds session and was an obvious presence on wards. Aims of the role were to:<br><br>a. Provide a central point of contact for health care providers to make bookings.<br><br>b. Coordinate the efficient use of on-site interpreters (i.e. pro-actively seeking clients who need a same-language interpreter and informing the medical team that they are on site).<br><br>c. Ensure the rostered interpreter is used effectively | 60 minute hospital-based training sessions provided by the Aboriginal Interpreter Service for all Gove District Hospital staff as well as Royal Darwin Hospital new interns during their orientation days, and for doctors in specific divisions (Emergency department; Surgical division), addressing: an introduction to how different languages work; overview of Aboriginal languages spoken in the Northern Territory; why context is important in communication; how to avoid common areas of miscommunication; how to communicate in plain English; how to work with an interpreter effectively; practical tips for booking and using Aboriginal interpreters. | medical officers working in the hospital participated as 'clinical champions', ensuring use of interpreters in their clinical role and advocating use to colleagues. 'Champions' met regularly with the study team to discuss barriers and facilitators, and ways to advocate for and promote the use of interpreters in their daily work. |
| Materials | posters and fliers alerting healthcare providers to the existence of the NT Aboriginal Interpreter Service and providing bookings information were displayed on hospital noticeboards. | Training materials including language names and maps were provided during the 'Working with Aboriginal Interpreter' training sessions. | None |
| Who provided | • Aboriginal interpreter coordinator employed by the hospital | • Interpreters employed by the NT Aboriginal Interpreter Service<br><br>• Clinical champion<br><br>• Study staff (Aboriginal and non-Aboriginal project staff and investigators) from Menzies School of Health Research | • Healthcare providers employed by the hospital |
| How | • Aboriginal leadership<br><br>○ was provided through the Aboriginal Interpreter Coordinator role itself (and management of and advocacy for that role through the Royal Darwin Hospital Aboriginal Support Unit and the Aboriginal and Torres Strait Islander Workers Advisory Group respectively), and through the management team of the Aboriginal Interpreter Service. Aboriginal researchers participated in qualitative and survey data collection on patient experience (reported separately).<br><br>• The Aboriginal Interpreter Coordinator:<br><br>○ Did ward rounds of the hospital to identify language needs and coordinate interpreter bookings was an obvious presence on wards.<br><br>○ Provided in-services on a regular basis to healthcare providers especially nurses and allied health practitioners, promoting interpreter use<br><br>○ Provided mentoring and support for onsite interpreters, especially the rostered interpreter | • 'Working with Interpreters' training sessions provided training as described above<br><br>○ Dates: June 2018, September 2018, October 2018, January 2019 | • Clinical championing of interpreter use was embedded into practice as described above.<br><br>• Regular meetings with Aboriginal Interpreter Coordinator, stakeholders and clinical champions were held<br><br>• Reports on study processes and a final report were communicated to hospital governance |

*(Continued)*

**Table 2.** (Continued)

| Item | Description | | |
|---|---|---|---|
| Where | Royal Darwin Hospital, Northern Territory, Australia. This is a 360-bed public tertiary referral hospital and the largest hospital in the Northern Territory. | Royal Darwin Hospital and Gove District Hospital. Gove District Hospital is a 30-bed remote public acute care facility in Northeast Arnhem Land. | Royal Darwin Hospital (3 champions) and Gove District Hospital (1 champion) |
| How much | One full time Aboriginal Interpreter Coordinator role was filled for the 12-month intervention period | Six 'Working with Interpreter' training sessions were held with 127 attendees; three clinical champions participated for the 12-month intervention period. | Three clinical champions participated |
| Tailoring | None | | |
| Modifications | Key roles of the Aboriginal interpreter coordinator evolved from the above-stated aims and ultimately focused chiefly on: 1. Advocacy: Advocating for the cultural and language needs of patients to improve communication and achieve culture change. 2 Mentoring and support for interpreters when on site: ensuring that interpreters, especially new staff, can be supported to feel confident in the challenging and potentially alienating healthcare environment. 3 Education for healthcare providers: on when and how to use an Aboriginal interpreter. | | |
| How well (reach) | The Aboriginal Interpreter Coordinator was active across all wards of the hospital (good reach) | Only a small proportion of total healthcare providers attended the working with interpreter training sessions (limited reach) | Only 3 clinical champions were involved in the study (limited reach) |

survey, 93 (73%) did the immediate post-training survey and 21 (17%) responded to the 6–8 month follow up survey, although not all survey questions were answered on each occasion (Table 3). Immediately post training, most participants (93%) reported the training was either fairly or very engaging and positive. Almost all (96%) were satisfied or very satisfied with the training. Confidence in ascertaining need for an interpreter increased from 24/88 (27.3%) to 52/93 (55.9%) of respondents indicating they were confident or very confident. After six to eight months, all respondents still felt fairly or very clear on the bookings process. Most (71%) felt they could ascertain the need for an interpreter more effectively than before the training and most (86%) indicated that the training had increased their likelihood of booking an interpreter. Improvements were noted in self-reported frequency of Aboriginal interpreter use ($\chi2 = 27.50$, $p < 0.05$), confidence ascertaining the need for an Aboriginal interpreter ($\chi2 = 26.02$, $p < 0.05$) and confidence in working with interpreters, at the two follow up surveys compared with the pre-training survey data ($\chi2 = 25.86$, $p < 0.05$).

## Intervention component 3: Clinical championing of interpreter use

Clinical championing of interpreter use comprised junior medical officers working in the hospital who ensured use of interpreters in their clinical practice where appropriate and advocated use to colleagues. A number of hospital clinicians across all levels of seniority have been long-standing champions for interpreter use and cultural safety in addition to those we specifically worked with during this study. For this study, champions met regularly with the study team to discuss barriers and facilitators, and ways to promote Aboriginal interpreter use in their daily work. This intervention component was incorporated since championing by peers or organisational leaders is considered an important strategy to achieve sustainable clinician behaviour change [22]. However the champions in this study were not in leadership positions. As junior

**Table 3. Comparison of clinicians clarity on the interpreter booking process immediately after the 'Working with Interpreters' training session and at six to eight months follow up.**

| Survey question | Pre-training | Post-training | Six to eight months follow-up |
|---|---|---|---|
| **Understanding of how to book an Aboriginal interpreter** | *This question not included* | n = 92 | n = 21 |
| Very clear, I could book one today if I needed to | | 32 (34.8) | 13 (61.9) |
| Fairly clear, I could find out how to do it reasonably easily | | 55 (59.8) | 8 (38.1) |
| Not very clear, it would take time and effort to find out | | 5 (5.4) | 0 (0) |
| Unclear, I wouldn't know where to start | | 0 (0) | 0 (0) |
| **Frequency of use of Aboriginal interpreters** | **n = 87** | **n = 91** | **n = 19** |
| Never | 38 (43.7) | 1 (1.1) | 1 (5.3) |
| Occasionally | 44 (50.6) | 22 (24.2) | 9 (47.4) |
| Regularly | 5 (5.7) | 64 (70.3) | 5 (26.3) |
| Always | 0 (0) | 4 (4.4) | 4 (21.1) |
| **Confidence ascertaining the need for an Aboriginal interpreter** | **n = 88** | **n = 93** | **n = 21** |
| Not at all confident | 1 (1.1) | 1 (1.1) | 0 (0) |
| Not confident | 14 (15.9) | 2 (2.2) | 0 (0) |
| Somewhat confident | 49 (55.7) | 38 (40.9) | 7 (33.3) |
| Confident | 23 (26.1) | 46 (49.5) | 10 (47.6) |
| Very confident | 1 (1.1) | 6 (6.5) | 4 (19) |
| **Confidence in working with an Aboriginal interpreter** | **n = 86** | **n = 92** | **n = 19** |
| Not at all confident | 3 (3.5) | 1 (1.1) | 0 (0) |
| Not confident | 9 (10.5) | 1 (1.1) | 0 (0) |
| Somewhat confident | 40 (46.5) | 27 (29.3) | 2 (10.5) |
| Confident | 29 (33.7) | 49 (53.3) | 12 (63.2) |
| Very confident | 5 (5.8) | 14 (15.2) | 5 (26.3) |

medical officers, they rotated through different rosters including night shifts. They had power to change their own practice and potentially influence colleagues, but since only three champions were engaged, their ability to impact a large health service was considered to be relatively low.

## Relationship between self-discharge and interpreter bookings

We identified that during the whole 3-year study period (baseline and intervention phases), a statistically significant inverse association was present between interpreter bookings and likelihood of self-discharge among Aboriginal inpatients (Fig 2); β coefficient -0.0078 (standard error 0.0028), p = 0.019, $R^2$ 0.4260 (S1 File).

## Other activities during 2016–2019

Independent from this study, an initiative in the hospital's cardiac care unit from January 2018 comprised production and distribution of Aboriginal language lanyard cards matching community names with languages to help staff determine what language the patient might speak. The cards also provide the telephone and email contacts for the Aboriginal Interpreter Service [23]. This initiative might have contributed to improved interpreter uptake, especially in the cardiac care unit. It commenced during the study baseline period but may have gained momentum during the intervention phase.

Additional initiatives aiming to decrease self-discharge and improve patient-centred care and cultural safety had been occurring, guided by TEHS's Organisational Culture Charter, the 2016 Northern Territory Aboriginal Cultural Security Framework [24] and the 2017

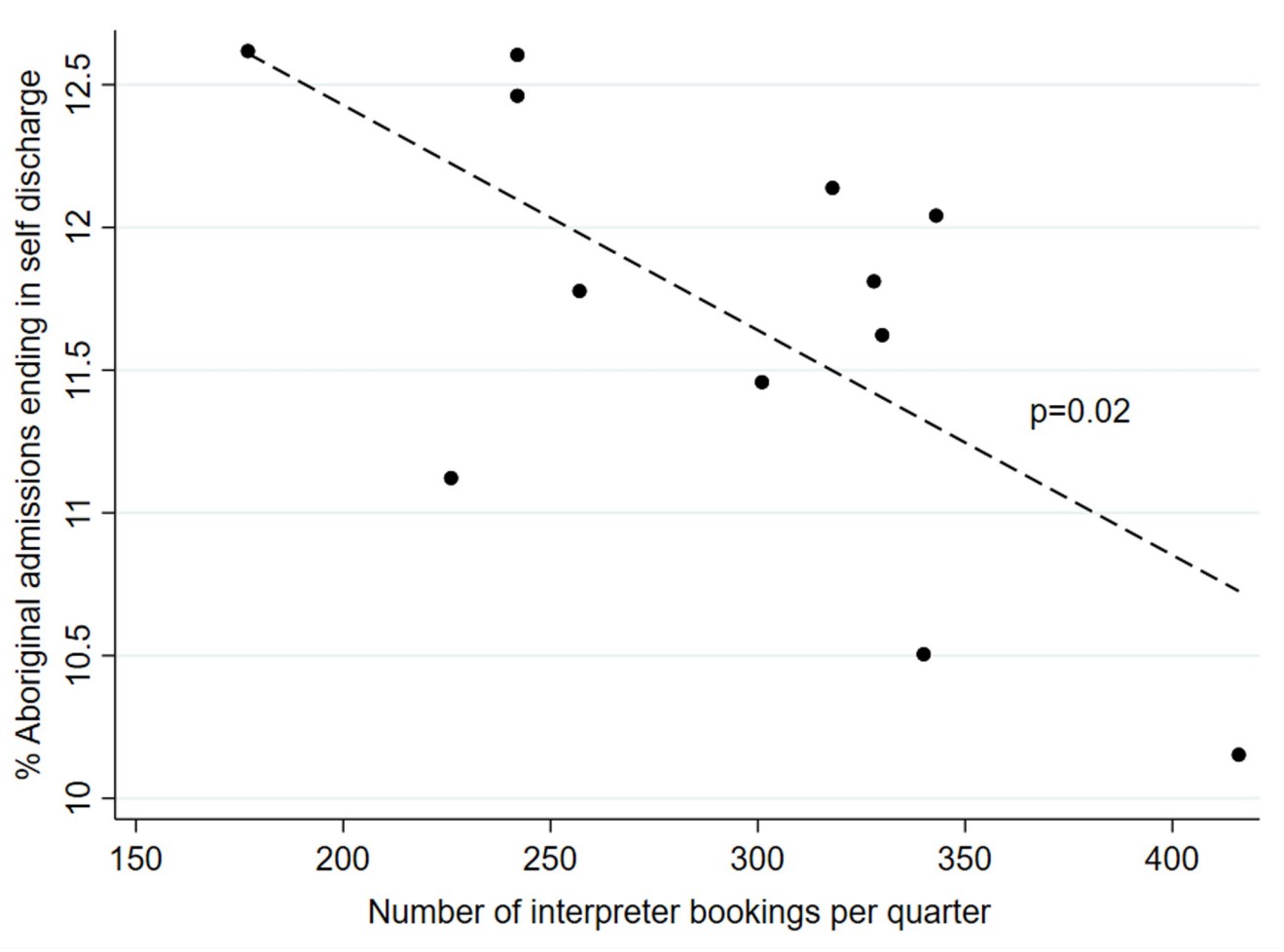

**Fig 2. Self discharge rates among Aboriginal inpatients according to number of interpreter bookings made per quarter during baseline and study intervention period (April 2016 to March 2019).** Dashed line shows line of best fit. P value derived from linear regression analysis.

Australian National Safety and Quality Health Service Standards [25]. The hospital foyer was renovated to position the Aboriginal Support Unit at the entrance, display Aboriginal artworks and acknowledge traditional owners (works completed December 2017). Additionally, in 2016 self-discharge data were included as a key indicator in monthly performance reports to motivate reductions in self-discharge rates. These activities may be contributing to growing momentum in provision of more culturally safe care (including increased interpreter use for Aboriginal language speakers) that improves patient experience and thereby decreases self-discharge. However, the activities clearly pre-dated the study intervention period. It is difficult to determine the extent to which these may have contributed to the specific changes shown in the interrupted time-series analysis.

## Discussion

During a hospital-based multi-component intervention to improve Aboriginal interpreter uptake, a significant increase in the proportion of Aboriginal patients gaining access to an interpreter occurred, and self-discharge rates among Aboriginal people fell from 12 to 10%. This equates to approximately 220 individuals avoiding self-discharge in one year; a modest

number compared to the >10,000 admissions annually, but of high clinical importance for those individuals. The timing of study outcomes illustrated in Fig 1 and detailed in the accompanying paper [12] coincided most obviously with the time of employment of the Aboriginal Interpreter Coordinator. The coordinator worked throughout all wards of the hospital and was employed fulltime to undertake the task of facilitating interpreter bookings, therefore this is most likely the single most important contributor to the positive study outcome obtained. Although the Working with Interpreter training was well received by participants and is likely to have also contributed to increased interpreter bookings, the number of participants in training was small relative to the overall number of healthcare providers employed at RDH, therefore reach was limited. The impact of the three junior clinical champions we considered likely to be small, based on the small number of champions, their junior position in the hospital hierarchy, and competing priorities. In teasing apart the potential components however we acknowledge that there may have been synergism from the 'bundled' interventions; multi-component strategies are acknowledged as being potentially more successful in changing health systems and clinician behaviour than single-strategy approaches [11, 26].

Previous research has identified that interventions seeking to 'restructure and reinforce new practice norms and associate them with peer and reference group behaviours' may be successful in achieving behaviour change [26]. In this study, the activities were intended to make it easier for clinicians to recognise the need for Aboriginal interpreters, book an interpreter more easily, and interact with the interpreter and patient more effectively–and to 'normalise' this behaviour. Approaching this through training and championing, as well as providing the mechanism to enable new behaviours (through a Coordinator) was associated with success in achieving the study's goals of increased interpreter uptake and improved patient outcome, measured as a decrease in self-discharge rates. The 'Working with Interpreters' training was very well received by clinicians and, along with better opportunities for Aboriginal cultural training which have recently been developed [27], should be incorporated routinely into the health service's staff training curricula.

The association between interpreter bookings and self-discharge rates was explored in more detail. The logic is that better-informed patients who have had access to an interpreter have a better comprehension of the need to receive care in hospital, and better experience of care, leading to a lesser likelihood of self-discharge. Not only did interpreter uptake and self-discharge show significant changes in gradient during the 12-month study implementation phase compared with the baseline period as already described [12]; these outcomes also showed a linear (inverse) relationship with each other throughout the 3-year baseline and intervention period (Fig 2 and S1 File). That interpreter bookings showed an association with self-discharge rates during the whole data collection period provides internal validity in attributing the fall in self-discharge rates to the rise in interpreter bookings. This is to our knowledge the first time this association has been shown, and corroborates previous suggestions of a likely association between better communication and lower self-discharge rates [8–10]. While this lends weight to there being a causal relationship between these measures, data linked by patient identifier would provide much stronger evidence for a direct association between access to an interpreter and reduction in self-discharge; this needs to be explored in future research.

Non-study activities may have contributed to the changes seen. The hospital has been striving to decrease self-discharge, currently targeting 7% (3% lower than was attained by the end of this study). New measures to improve cultural safety for Aboriginal people have been implemented. However, the study intervention dates showed a particular association with the change in outcomes.

A key limitation of this study is the difficultly in accurately measuring 'reach' of each intervention component. Qualitative data exploring why clinicians do or do not use interpreters

may have helped to further determine the relative importance of different study components. Related in-depth qualitative work is underway focusing on remaining barriers [16] and further system-strengthening activities are planned to scale up these study findings and to implement the Australian National Standards on Quality and Safety in Health care, which include 'communicating for safety' [25]. Survey data from training participants provided only subjective self report and are limited by response bias which may have favoured more positive attitudes to the training; but other research from the same health service using survey data achieved a high response rate and also found that staff appreciate and want more training about how to interact effectively with Aboriginal patients [28], corroborating our findings here. A further potential limitation is that we have used TIDieR as an evaluation framework whereas it was developed as a descriptive tool for intervention components [14]. However, we suggest it has a useful role in this regard due to the clarity and simplicity of the structure.

## Conclusions

This study demonstrates that beneficial patient outcomes can be achieved through health service systems changes that dedicate additional resourcing to identified areas of need. Use of Aboriginal language interpreters for people who primarily speak an Aboriginal language is one critically important component of effective communication in the provision of healthcare. Improvements in interpreter uptake, with associated change in health behaviours and outcomes, can be achieved through systems changes incorporating Aboriginal leadership. Further strategies to escalate the proportion of Aboriginal patients getting access to high-quality communication in this setting are required as a core strategy to improve health outcomes. To achieve the much greater-magnitude change now required, substantial investment in combined approaches for upscaling Aboriginal interpreter use addressing supply, demand, efficiency and effectiveness, are needed.

## Supporting information

**S1 File. Regression analysis.**
(DOCX)

## Acknowledgments

We greatly thank the Aboriginal Interpreter Service and Top End Health Services for provision of data.

## Author Contributions

**Conceptualization:** Elise O'Connor, Robyn Aitken, Craig Castillon, Curtis Roman, Anna P. Ralph.

**Data curation:** Elise O'Connor, Anna P. Ralph.

**Formal analysis:** Elise O'Connor, Vicki Kerrigan, Anna P. Ralph.

**Funding acquisition:** Anna P. Ralph.

**Investigation:** Elise O'Connor, Craig Castillon, Vincent Mithen, Anna P. Ralph.

**Methodology:** Elise O'Connor, Vicki Kerrigan, Robyn Aitken, Craig Castillon, Curtis Roman, Anna P. Ralph.

**Project administration:** Vincent Mithen.

**Resources:** Craig Castillon, Vincent Mithen, Gail Madrill, Curtis Roman, Anna P. Ralph.

**Supervision:** Robyn Aitken, Gail Madrill.

**Validation:** Vincent Mithen.

**Writing – original draft:** Elise O'Connor, Vicki Kerrigan, Anna P. Ralph.

**Writing – review & editing:** Elise O'Connor, Vicki Kerrigan, Robyn Aitken, Vincent Mithen, Gail Madrill, Curtis Roman, Anna P. Ralph.

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
