## [Decision Letter · Decision Letter 0]

20 Apr 2021

PONE-D-21-06717

Does improved interpreter uptake reduce self-discharge rates in hospitalised patients? A successful hospital intervention explained

PLOS ONE

Dear Dr. Ralph,

Thank you for submitting your manuscript to PLOS ONE. After careful consideration, we feel that it has merit but does not fully meet PLOS ONE’s publication criteria as it currently stands. Therefore, we invite you to submit a revised version of the manuscript that addresses the points raised during the review process.

We look forward to receiving your revised manuscript.

Kind regards,

Vijayaprakash Suppiah, PhD

Academic Editor

PLOS ONE

Journal Requirements:

3. We noted in your submission details that a portion of your manuscript may have been presented or published elsewhere.

[We submitted an earlier version of this paper to Implementation Science Communications. A pre-print was generated which remains accessible here: doi 10.21203/rs.3.rs-62995/v1 . The manuscript we are submitting to PLOS ONE is revised with a new title, revised abstract and inclusion of additional data. We have communicated with Mark Brewin at Research Square Preprint Plaform requesting that the preprint be removed This request has been denied. He states that the presence of a preprint should not impact on subsequent submission to another journal.]

Please clarify whether this publication was peer-reviewed and formally published. If this work was previously peer-reviewed and published, in the cover letter please provide the reason that this work does not constitute dual publication and should be included in the current manuscript.

Reviewers' comments:

Reviewer's Responses to Questions

**Comments to the Author**

1. Is the manuscript technically sound, and do the data support the conclusions?

Reviewer #1: Yes

Reviewer #2: Yes

2. Has the statistical analysis been performed appropriately and rigorously? 

Reviewer #1: Yes

Reviewer #2: No

3. Have the authors made all data underlying the findings in their manuscript fully available?

Reviewer #1: Yes

Reviewer #2: No

4. Is the manuscript presented in an intelligible fashion and written in standard English?

Reviewer #1: Yes

Reviewer #2: Yes

5. Review Comments to the Author

Reviewer #1: Does improved interpreter uptake reduce self-discharge rates in hospitalised patients? A successful hospital intervention explained

• In the data and definitions section ;

RDH 77 separations data, used as a measure of inpatient healthcare utilisation, were obtained for all 78 Indigenous people for the same timeframe.’

Previously you have used Aboriginal. Is Indigenous used here because of the hospitals own coding preference ? Otherwise I would suggest a consistent term be used throughout

• Could you make it more clear who the falls into the category of ‘clinicians’ ? is it doctors only or nurses and allied health also – if these parties were not trained is there a reason?

• Why was only surgical and emergency doctors trained , rather than other wards or specialities?

• The cultural safety paragraph is well written , clear and well referenced.

In this section you have included explanation on your terminology but I wonder if this concept would be better introduced earlier.

• ‘Additionally, implementation and delivery included Aboriginal leadership.’ Could you expand on this – who was consulted for the development of the training or the training of the interpreter coordinator, were these hospital employees? Doctors? Community members?

• Intervention component 3- you mentioned meeting with the champions about the barriers and facilitators – could you expand on what these were?

• ‘While the role was conceived as being a coordinator role to improve efficiency and ease of bookings, it was in fact realised somewhat differently, focussing more on Aboriginal staff support particularly, provision of mentoring for interpreters on assignment to the hospital, including helping them to navigate the hospital environment.’ This is interesting, I would like to know more about the training of the interpreters themselves, perhaps it is out of the scope of this article.

• It is good that you reported on the additional interventions occurring in the study period. It may be devaluing and overzealous to say that of these initiatives that it is ‘unlikely to have had a large role’ – perhaps alter your wording here. Also in regards to intern orientation – was there any other changes with cultural safety training from year to year – potentially the cultural safety knowledge overall was increased hence the appreciation of the interpreter service.

• The first paragraph of your discussion is very well worded, and very clear.

• Overall this paper is very clear and very well written.

Your table outlines some of the more specific details that I have questioned above but consider if there is scope to have some of your explanations expanded in the main body of the text. You’re reporting on the self discharge numbers and associations with interpreter use is quiet brief however I see this as the main focus of the paper – I am not sure if there is any date or scope to expand on this.

Otherwise I wonder if you could make any recommendations in your discussion about how this intervention might be applicable to other hospital or health service settings. I appreciate you’re suggestions for further research in this area.

Well done. I hope these interventions can continue to the be the ‘norm’ in the hospital.

Reviewer #2: This is an interesting well written article that addresses an important and concerning issue for healthcare providers.

There are a couple of points I would like to make:

Line 94- Evaluation and analysis section – state significance level used in study

Line 155. After noting satisfaction in training immediately after the initial training it would be useful to know the degree of increased confidence participants had regarding interpreters in text (as noted in follow up), although I note it is included in the table but it is useful to read about the immediate effect of training.

Line 180 – relationship between self-discharge and interpreter bookings. State beta coefficient and p value. I note in the journal information that for regression analyses it is necessary to include the full results of any regression analysis performed as a supplementary file. Include all estimated regression coefficients, their standard error, p-values, and confidence intervals, as well as the measures of goodness of fit.

Line 213 – the authors note that there was a decline in self-discharge from 12-10%. It may be helpful to highlight how many people are we talking about and the clinical significance of this

Line 219 - the authors state that the number of people who did workshops was low compared to numbers of healthcare professionals employed. Could the numbers included in the results section (line 147-)

Check References are Vancouver style

Figure 2 – is it possible to include more data points to help with interpretation

6. PLOS authors have the option to publish the peer review history of their article (what does this mean?). If published, this will include your full peer review and any attached files.

Reviewer #1: No

Reviewer #2: No

---

## [Author Response · Author response to Decision Letter 0]

16 May 2021

29 April 2021

Dear Editor,

Re: Response to reviewer comments

PONE-D-21-06717: Does improved interpreter uptake reduce self-discharge rates in hospitalised patients who speak diverse languages? A successful hospital intervention explained 

We thank both reviewers for the positive and helpful comments. Please find our responses in bold in the attached letter, and pasted unformatted below. 

Reviewer #1: Does improved interpreter uptake reduce self-discharge rates in hospitalised patients? A successful hospital intervention explained

• In the data and definitions section ;

RDH separations data, used as a measure of inpatient healthcare utilisation, were obtained for all Indigenous people for the same timeframe.’

Previously you have used Aboriginal. Is Indigenous used here because of the hospitals own coding preference ? Otherwise I would suggest a consistent term be used throughout

and 

• The cultural safety paragraph is well written , clear and well referenced.

In this section you have included explanation on your terminology but I wonder if this concept would be better introduced earlier.

To respond to these two points, we have now used ‘Aboriginal and Torres Strait Islander peoples’ instead of Indigenous in line 78 (in line 83 we then explain that Torres Strait Islanders were excluded) and also in line 114. Therefore we no longer need to use the term Indigenous except where it appears in the title of another document (line 113) or as stated in another document (p 117). We have therefore taken out the definition oof Indigenous and use Aboriginal predominantly, reflecting the population of focus in this study.

• Could you make it more clear who the falls into the category of ‘clinicians’ ? is it doctors only or nurses and allied health also – if these parties were not trained is there a reason?

We had added “Additionally, three training sessions were conducted for regional hospital staff in June 2018 (mostly doctors plus 1 nurse; Table 2)”. 

• Why was only surgical and emergency doctors trained , rather than other wards or specialities? 

We have added “Surgery and Emergency are high-priority areas for interpreter use. Decisions about where to target training were also based on pragmatic factors, such as which teams had existing teaching rosters that could be utilised.”

• ‘Additionally, implementation and delivery included Aboriginal leadership.’ Could you expand on this – who was consulted for the development of the training or the training of the interpreter coordinator, were these hospital employees? Doctors? Community members? In particular this refers to the role of the Aboriginal Interpreter Coordinator 

• Intervention component 3- you mentioned meeting with the champions about the barriers and facilitators – could you expand on what these were?

• ‘While the role was conceived as being a coordinator role to improve efficiency and ease of bookings, it was in fact realised somewhat differently, focussing more on Aboriginal staff support particularly, provision of mentoring for interpreters on assignment to the hospital, including helping them to navigate the hospital environment.’ This is interesting, I would like to know more about the training of the interpreters themselves, perhaps it is out of the scope of this article.

Indeed this is a critical issue but beyond scope for this paper. Our follow up project plans to ensure that Aboriginal interpreters receive more training in health concepts and terminology. The Northern Territory Primary Health Network is working with the Aboriginal interpreter Service to develop a Plain English Health Dictionary which will support this. We are now partnering with the National Accreditation Authority for Translators and Interpreters with the goal of getting more Aboriginal interpreters certified at level 3 (the highest level of attainment). 

• It is good that you reported on the additional interventions occurring in the study period. It may be devaluing and overzealous to say that of these initiatives that it is ‘unlikely to have had a large role’ – perhaps alter your wording here. Also in regards to intern orientation – was there any other changes with cultural safety training from year to year – potentially the cultural safety knowledge overall was increased hence the appreciation of the interpreter service.

We have changed this section to read: “However, the activities clearly pre-dated the study intervention period. It is difficult to determine the extent to which these may have contributed to the specific changes shown in the interrupted time-series analysis.”

Your table outlines some of the more specific details that I have questioned above but consider if there is scope to have some of your explanations expanded in the main body of the text. Your reporting on the self discharge numbers and associations with interpreter use is quiet brief however I see this as the main focus of the paper – I am not sure if there is any date or scope to expand on this. We have now included more of the information from tables in the text (see also responses to Reviewer 2) and have included a new supplementary file detailing the relationship between self discharge rates and interpreter bookings. 

Otherwise I wonder if you could make any recommendations in your discussion about how this intervention might be applicable to other hospital or health service settings. I appreciate your suggestions for further research in this area.

We have added in the Conclusion: “This study demonstrates that beneficial patient outcomes can be achieved through health service systems changes that dedicate additional resourcing to identified areas of need.”

Reviewer #2: This is an interesting well written article that addresses an important and concerning issue for healthcare providers.

Line 94- Evaluation and analysis section – state significance level used in study We have added “Statistical significance was defined as a p value of less than 0.05.”

Line 155. After noting satisfaction in training immediately after the initial training it would be useful to know the degree of increased confidence participants had regarding interpreters in text (as noted in follow up), although I note it is included in the table but it is useful to read about the immediate effect of training. We have added in new line 163: “Confidence in ascertaining need for an interpreter increased from 24/88 (27.3%) to 52/93 (55.9%) of respondents indicating they were confident or very confident.”

Line 180 – relationship between self-discharge and interpreter bookings. State beta coefficient and p value. I note in the journal information that for regression analyses it is necessary to include the full results of any regression analysis performed as a supplementary file. Include all estimated regression coefficients, their standard error, p-values, and confidence intervals, as well as the measures of goodness of fit.

We now include the following sentence in the results: ‘We identified that during the whole 3-year study period (baseline and intervention phases), a statistically significant inverse association was present between interpreter bookings and likelihood of self-discharge among Aboriginal inpatients (Fig 2); � coefficient -0.0078 (standard error 0.0028), p=0.019, R2 0.4260 (S1 file).’

We also include the following as a supplementary file.

Linear regression analysis was undertaken in Stata 15.1 to the examine the relationship between quarterly numbers of Aboriginal interpreter bookings and reported episodes of care ending discharge for Aboriginal patients. The ‘robust’ option was used to control for heteroskedasticity.

pct_tol: percentage of Aboriginal patients who took own leave (self discharged / discharged against medical advice) per quarter

quarterly_interp_bookings: number of Aboriginal interpreter bookings made per quarter

The beta coefficient was -0.0078

The p value was 0.019

The goodness of fit as indicated by R squared was 0.4260

Post-estimation diagnostics using predicted pct_tol plotted against actual pct_tol revealed an approximately directly proportional relationship. We plotted residuals against predicted values and did not observe any pattern indicating appropriateness of the model.

regress pct_tol quarterly_interp_bookings, robust

Linear regression Number of obs = 12

 F(1, 10) = 7.74

 Prob > F = 0.0194

 R-squared = 0.4260

 Root MSE = .62421

 | Robust

 pct_tol | Coef. Std. Err. t P>|t| [95% Conf. Interval]

+----------------------------------------------------------------

quarterly_interp_bookings | -.0078402 .0028187 -2.78 0.019 -.0141207 -.0015598

 _cons | 13.9885 .8335761 16.78 0.000 12.13118 15.84582

Line 213 – the authors note that there was a decline in self-discharge from 12-10%. It may be helpful to highlight how many people are we talking about and the clinical significance of this. We have added in the methods: “After applying these exclusions, there were 21,633 separations among Aboriginal people in the two-year baseline and 10,919 in the 1-year intervention period [12].” We have added in the first paragraph of the discussion after mentioning the 12-10% decrease: “This equates to approximately 220 individuals avoiding self-discharge in one year; a modest number compared to the >10,000 admissions annually, but of high clinical importance for those individuals.”

Line 219 - the authors state that the number of people who did workshops was low compared to numbers of healthcare professionals employed. Could the numbers included in the results section (line 147-) We have added in line 151: “‘Working with Interpreters’ training sessions were provided by the Aboriginal Interpreter Service in TEHS hospitals to 127 participants.” Unfortunately we have been unable to obtain a denominator (number of clinical employees at that time), but we estimate 600-1000 healthcare providers are employed by the health service or work as locum staff at the health service. 

Check References are Vancouver style Yes, references are Vancouver style and correct for PLOS; except that the PubMed Central PMCID number is appearing even though the bibliography template indicates it has not been added. I will manually remove these if required by the editor. 

Figure 2 – is it possible to include more data points to help with interpretation Unfortunately this is beyond our capacity. The statistician decided on quarterly rather than monthly bins and it has proven difficult to re-analyse. 

Editor’s query: Was the pre-print publication that is related to this one was peer-reviewed and formally published? No, we had submitted a previous shorter version of the paper (without all the data now included in this improved version) to Implementation Science. In this process it appeared as a pre-print with an assigned doi, however it was never peer reviewed and has never been accepted for publication elsewhere. We sought to have the pre-print taken down from the internet but were instructed that this was not possible. We note the PLOS ONE statement indicating that the existence of a pre-print version will not impact ability to publish in PLOS ONE. 

Please ensure that your manuscript meets PLOS ONE's style requirements. We have made changes to headings and the acknowledgement statement to conform with style requirements. 

We greatly thank the reviewers for their helpful comments which have improved manuscript. 

Yours sincerely,

Professor Anna Ralph (corresponding author)

---

## [Decision Letter · Decision Letter 1]

14 Jul 2021

PONE-D-21-06717R1

Does improved interpreter uptake reduce self-discharge rates in hospitalised patients? A successful hospital intervention explained

PLOS ONE

Dear Dr. Ralph,

Thank you for submitting your manuscript to PLOS ONE. After careful consideration, we feel that it has merit but does not fully meet PLOS ONE’s publication criteria as it currently stands. Therefore, we invite you to submit a revised version of the manuscript that addresses the points raised during the review process.

We look forward to receiving your revised manuscript.

Kind regards,

Vijayaprakash Suppiah, PhD

Academic Editor

PLOS ONE

Journal Requirements:

Additional Editor Comments (if provided):

Reviewers' comments:

Reviewer's Responses to Questions

**Comments to the Author**

1. If the authors have adequately addressed your comments raised in a previous round of review and you feel that this manuscript is now acceptable for publication, you may indicate that here to bypass the “Comments to the Author” section, enter your conflict of interest statement in the “Confidential to Editor” section, and submit your "Accept" recommendation.

Reviewer #1: All comments have been addressed

Reviewer #2: (No Response)

2. Is the manuscript technically sound, and do the data support the conclusions?

Reviewer #1: Yes

Reviewer #2: Partly

3. Has the statistical analysis been performed appropriately and rigorously? 

Reviewer #1: Yes

Reviewer #2: Yes

4. Have the authors made all data underlying the findings in their manuscript fully available?

Reviewer #1: Yes

Reviewer #2: Yes

5. Is the manuscript presented in an intelligible fashion and written in standard English?

Reviewer #1: Yes

Reviewer #2: Yes

6. Review Comments to the Author

Reviewer #1: Thank you for your considered and detailed responses, the additional information provided and included will allow a better ease of readership and maintains focus on the titled issue of interpreters and impact on self discharge. I would encourage you to publish your future findings about interpreter training and support you have mentioned. Well done in your manuscript.

Reviewer #2: although all responses to reviewer 2 have been addressed, I have a couple of comments about the authors' responses to reviewer 1's questions.

1. Line 158, authors have added "mostly doctors, plus one nurse". - are you able to specify the number of doctors

2. line 120, authors have responded to query about "aboriginal leadership" in their response to the reviewer but not changed anything in text - can you change to state that the Aboriginal Interpreter coordinator was involved in the implementation and delivery of the program - this removes any ambiguity around this

7. PLOS authors have the option to publish the peer review history of their article (what does this mean?). If published, this will include your full peer review and any attached files.

Reviewer #1: No

Reviewer #2: No

---

## [Author Response · Author response to Decision Letter 1]

28 Aug 2021

16 July 2021

Dear Editor,

Re: Second response to reviewer comments

PONE-D-21-06717 R1: Does improved interpreter uptake reduce self-discharge rates in hospitalised patients who speak diverse languages? A successful hospital intervention explained 

We thank both reviewers for re-reviewing. Please find our responses in bold below. 

Reviewer #1: Thank you for your considered and detailed responses, the additional information provided and included will allow a better ease of readership and maintains focus on the titled issue of interpreters and impact on self discharge. I would encourage you to publish your future findings about interpreter training and support you have mentioned. Well done in your manuscript.

Reviewer #2: although all responses to reviewer 2 have been addressed, I have a couple of comments about the authors' responses to reviewer 1's questions.

1. Line 158, authors have added "mostly doctors, plus one nurse". - are you able to specify the number of doctors

The text has now been updated to include the number provided in table 2: 126 doctors and 1 nurse. 

2. line 120, authors have responded to query about "aboriginal leadership" in their response to the reviewer but not changed anything in text - can you change to state that the Aboriginal Interpreter coordinator was involved in the implementation and delivery of the program - this removes any ambiguity around this

We have revised this to read: ‘Additionally, implementation and delivery included Aboriginal leadership, provided through the Aboriginal Interpreter Coordinator, the Royal Darwin Hospital Aboriginal Support Unit and the Aboriginal Interpreter Service.’ 

This ensures all three Aboriginal authors who contributed to implementation and delivery of the study are adequately acknowledged. 

We greatly thank the reviewers for their helpful comments which have improved manuscript. 

Yours sincerely,

Professor Anna Ralph (corresponding author)

---

## [Editor Report · Decision Letter 2]

13 Sep 2021

Does improved interpreter uptake reduce self-discharge rates in hospitalised patients? A successful hospital intervention explained

PONE-D-21-06717R2

Dear Dr. Ralph,

We’re pleased to inform you that your manuscript has been judged scientifically suitable for publication and will be formally accepted for publication once it meets all outstanding technical requirements.

Kind regards,

Vijayaprakash Suppiah, PhD

Academic Editor

PLOS ONE

---

## [Editor Report · Acceptance letter]

30 Sep 2021

PONE-D-21-06717R2 

Does improved interpreter uptake reduce self-discharge rates in hospitalised patients? A successful hospital intervention explained 

Dear Dr. Ralph:

I'm pleased to inform you that your manuscript has been deemed suitable for publication in PLOS ONE. Congratulations! Your manuscript is now with our production department. 

Kind regards, 

on behalf of

Dr. Vijayaprakash Suppiah 

Academic Editor

PLOS ONE